# Comparison of clinical practice guidelines methods to reach diagnostic test recommendations regarding diagnostic laparoscopy for endometriosis: A scoping review

**Frank Zela-Coila**[1☯], **Carlos Quispe-Vicuña**[2☯], **Janeth N. Nuñez-Lupaca**[3☯], **Milagros Aparicio-Curazi**[1☯], **Sergio Goicochea-Lugo**[4,5☯]*

1 Sociedad Científica de Estudiantes de Medicina Agustinos, Universidad Nacional de San Agustín de Arequipa, Arequipa, Peru, 2 Sociedad Científica de San Fernando, Lima, Peru, 3 Universidad Nacional Jorge Basadre Grohmann, Tacna, Peru, 4 Unidad de Investigación para la Generación y Síntesis de Evidencias en Salud, Universidad San Ignacio de Loyola, Lima, Peru, 5 EviSalud—Evidencias en Salud, Lima, Peru

☯ These authors contributed equally to this work.
* sgoicochealugo@gmail.com

## Abstract

### Introduction

Although Clinical Practice Guidelines (CPG) highlight that laparoscopy is often used in the treatment of endometriosis, its diagnostic usefulness is not fully defined. Our objective was to evaluate the quality of CPGs for endometriosis that address the use of diagnostic laparoscopy in reproductive age women, and describe the recommendations and methods used to assess diagnostic test questions.

### Methods

A comprehensive search of 5 databases (Trip Database, MEDLINE/PubMed, Web of Science, SCOPUS, and EMABSE) and websites of guideline development organizations and compilers was conducted from 2017 to 2023. A descriptive analysis of the recommendations was performed and the quality of the guidelines was assessed using the AGREE-II instrument.

### Results

Four CPGs were included in the review, all exhibiting adequate methodological quality (scores ranging from 66.7% to 91.0%). Regarding the use of laparoscopy for endometriosis diagnosis, discrepancies in recommendations were observed. Two guidelines advised against it, one recommended either laparoscopy or medical empirical treatment, and one favored its use. GRADE guidance was employed for evidence assessment, but only one

**Data Availability Statement:** All relevant data are within the manuscript and its Supporting Information files.

**Funding:** The author(s) received no specific funding for this work.

**Competing interests:** The authors have declared that no competing interests exist.

guideline transparently reported the certainty of evidence and the evidence-to-decision framework process.

## Conclusions

Variability in recommendations among different CPGs were found. To keep in mind, discrepancies arise from differing prioritizations of the assessment of clinical impact in patient important outcomes and methodological approaches. This underscores the need for more standardized and transparent guideline development processes, particularly in addressing the clinical utility of diagnostic tests.

## Introduction

Endometriosis affects 10% of reproductive-age women worldwide and is characterized by chronic pain and infertility [1–3]. However, diagnosing endometriosis remains challenging, involving a delay of 6 to 11.7 years [4–7]. While laparoscopy is considered the reference standard based on its results in diagnostic accuracy outcomes, it is important to consider the certainty of these results, the operator-dependent accuracy [8] and to keep in mind that the effects on clinically important outcomes for patients should also be known [9]. Additionally, laparoscopy is an invasive, expensive procedure, difficult to access in certain contexts, which is not devoid of complications, and there may be other alternatives to the use of laparoscopy as a diagnostic test that are non-invasive, such as empirical medical treatment [10, 11].

Clinical practice guidelines (CPGs) can address these types of uncertain scenarios by issuing recommendations based on systematic searches and appropriate methodological procedures to resolve questions about the use of diagnostic tests. In this regard, the Grading of Recommendations, Assessment, Development and Evaluation (GRADE) methodology proposes methodological guidelines for addressing this type of clinical questions, emphasizing decision-making on the clinical impact of diagnostic test such as laparoscopy on clinically important outcomes for patients, and considering a transparent multicriteria framework for issuing recommendations [12–16].

Prioritizing or not the decision-making process based on the clinical impact of the use of diagnostic tests, as well as employing different methods for issuing recommendations and properly adhering to them, can lead to contradictory recommendations among CPGs. In consequence, this study aims to evaluate the quality of CPGs for endometriosis that address the use of diagnostic laparoscopy, analyzing recommendations and methods to reach recommendations for this type of clinical question.

## Methods

We performed a scoping review following the methodology detailed by the Joanna Briggs Institute [17], following the guidelines of the Preferred Reporting Items for Systematic and Meta-Analysis extension for Scoping Reviews (PRISMA-ScR) [18] (**S1 Table**), and registered the study protocol in the repository Figshare [19].

### Eligibility criteria

We included only evidence-based clinical practice guidelines (CPG), defined as a document that formulates recommendations based on systematic reviews of the literature, that provide

recommendations regarding the use of laparoscopy for the diagnosis of pelvic endometriosis in reproductive age women with endometriosis, published or fully/partially updated during 2017 to 2023, and were available in English or Spanish language.

We excluded guidelines whose recommendations are adopted or adopted, and those whose methodology and recommendations cannot be obtained in full text. In addition, we excluded systematic reviews, observational studies, clinical trials, conference abstracts, letters to the editor, and case reports.

## Information sources and search strategy

We searched in databases (Trip Database, MEDLINE/PubMed, Web of Science, SCOPUS, and EMBASE), web pages of guideline development organizations (Scottish Intercollegiate Guidelines Network [SIGN], National Institute for Health and Care Excellence [NICE], Australian Clinical Practice Guidelines, New Zealand Guidelines Group, Centro Nacional de Excelencia Tecnológica en Salud [CENETEC], Guías Salud, Instituto de Evaluación de Tecnologías en Salud e Investigación [IETSI], and the American College of Physicians Clinical Practice Guidelines), and web pages of guideline compilers (Canadian Medical Association Infobase, Base Internacional de Guías GRADE, Guidelines International Network, National Guideline Clearinghouse, EGuideline, and Best Practice Guideline); from January 1st, 2017, to July 23th, 2023. Complete search strategy for each database in **S2 Table**.

## Study selection

We imported the articles from databases into Rayyan software, where duplicates were manually removed. Then, the obtained results were merge with articles sourced from guideline development organization web pages and guideline compilers, followed by a second manual removal of duplicates. Subsequently, four researchers (JNL, FZC, MAC, CQV) independently screened titles and abstracts to identify potentially relevant articles for inclusion. These potential articles were found and full text reviewed independently by the same researchers to verify compliance with the inclusion criteria. The disagreements were resolved with a fifth researcher (SGL).

## Data extraction

Four researchers (JNL,FZC,MAC,CQV) independently extract the following information from each included guideline: CPGs characteristics (organization, year, country, perspective, scope, method used to assess the certainty of the evidence and its definition, system to classify the strength of recommendations and its definition, and funding), characteristics regarding the recommendations on the use of laparoscopy for the diagnosis of endometriosis, and the methods used to assess diagnostic test clinical questions. In addition, we performed the quality appraisal of each CPG.

## Quality appraisal

We used the Appraisal of Guidelines Research and Evaluation II (AGREE-II) instrument [20], which includes 23 items distributed in six domains (scope and purpose, involvement of stakeholders, rigor of development, clarity, and presentation, applicability, and editorial independence) to assess the quality of CPGs. Three researchers (forming groups distributed between JNL, FZC, MAC, CQV) independently rated each item on a seven-point scale from strongly disagree (score 1) to strongly agree (score 7). When a difference of two or more points was found in one of the items between researchers, a fifth researcher (SGL) solved the

disagreement and reach a consensus. Then we followed the AGREE-II guidelines to calculate the global score for each domain ranging from 0 to 100%, where higher score indicates better quality.

The guideline of the instrument does not provide a minimum domain score to classify the quality of the guidelines. In consequence, we used a cut-off point based on previous studies that evaluated the quality of GPCs in gynecology and obstetrics topics [21, 22]. We considered that a CPG had an adequate quality when all six domains had a score $\geq$ 60%, and an adequate methodological quality when the third domain (rigor of development) had a score $\geq$ 60%.

### Appraisal of recommendations and methods to assess diagnostic test questions in CPGs

We extracted recommendations regarding the use of laparoscopy for the diagnosis of endometriosis from each CPG and described the following characteristics: recommendation (as state in the CPG), direction, strength, and certainty of the evidence. In addition, we described the methods used to reach the recommendations.

To our best knowledge, there is no specific validated tool for reporting and appraising methods to assess diagnostic test questions in CPGs. We utilized the GRADE guidelines as a reference standard methodology for formulating recommendations on this type of questions. GRADE was chosen due to its explicit steps guiding guideline developers in assessing the certainty of evidence for test accuracy, transitioning from test accuracy to patient-important outcomes, and making recommendations about diagnostic tests using multicriteria frameworks [12–16].

In based on that, we analyzed and described the following methodological characteristics: procedures to generate clinical question and decide on the importance of outcomes (clinical question, PICO framework, role of the diagnostic test, assessment of patient important outcomes or consequences of correct/incorrect classification), perform the systematic review of the evidence and determine the certainty of the evidence (databases and date of search, restrictions, type of study to be search, tool to assess risk of bias or quality of studies, method or criteria to determine certainty of evidence), prepare and report summaries of evidence tables (type of evidence table used, pre-test probability, principal results), and to discuss evidence to decision criteria (EtD table use, benefits, harms, certainty of the evidence, values and preferences, balance between benefits and harms, resource use or cost-effectiveness, equity, acceptability, feasibility).

### Data analysis and presentation

The synthesis of the results was descriptive. We used tables to present our findings.

## Results

### Study selection

We identified 4353 records from systematic search. After removed duplicates, 2479 records were screened by title and abstract, resulting in 54 records to evaluate full text for eligibility. We excluded 50 records after full text analysis and finally 4 CPGs were included. In addition, no other guideline was added from the citation searching of the included CPGs. **Fig 1** presents the flow diagram of CPG selection in detail and the list of excluded articles and reasons for exclusion is available in **S3 Table**.

The included CPG were: "Guía de práctica clínica para el diagnóstico y tratamiento de la endometriosis sintomática en mujeres en edad reproductiva" developed by IETSI (IETSI-

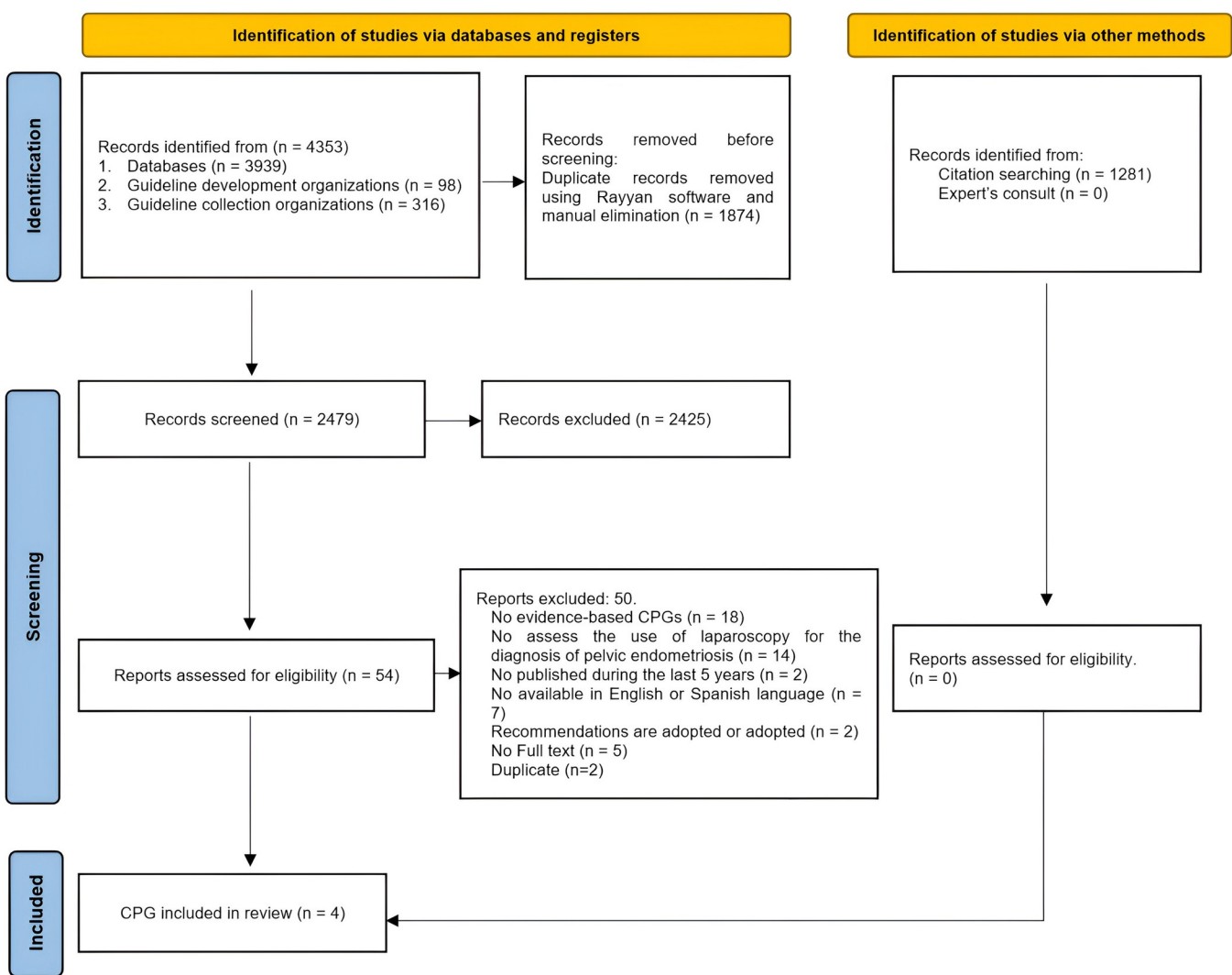

**Fig 1. PRISMA 2020 flow diagram for new systematic reviews which included searches of databases, registers and other sources.**

2022) [23], "Endometriosis" developed by European Society of Human Reproduction and Embryology (ESHRE-2022) [11], "Manejo de la paciente con endometriosis en la edad fértil" developed by Sociedad Española de Fertilidad (SEF-2018) [24], and Endometriosis: diagnosis and management" developed by NICE (NICE-2017) [25].

## Study characteristics

The CPG were developed in Peru [23], Spain [24], United Kingdom [25], and in a collaboration between European countries [11]. The perspective was not reported in three of four (3/4) guidelines [11, 24, 25], and one guideline used a population perspective [23]. The main scope for guideless was diagnosis and treatment. Two of four (2/4) guidelines were published in 2022 [11, 23]. All CPG used GRADE as the method to assess the certainty of the evidence and three of four (3/4) CPG used GRADE as the system to classify the strength of recommendations. Definitions for the certainty of the evidence, strength of recommendations and the terminology used to phrase the strength of the recommendations varied among the guidelines. In

addition, two guidelines were funded by public funds [23, 25], and the remaining ones did not report the source of funding [11, 24]. **Table 1** presents the characteristics and definitions used by CPG in detail.

## Quality appraisal of CPGs

Only one guideline (NICE-2017) [25] had an adequate quality in all domains of AGREE-II and all the CPG had an adequate methodological quality. In this sense, the CPG with the highest score was IETSI-2022 (91.0%) [23], while the one with the lowest score was SEF-2018 (66.7%) [24]. **Table 2.** Presents the quality appraisal of guidelines in detail.

## Synthesis of results

**Recommendations regarding the use of diagnostic laparoscopy.** Two guidelines state recommendations against the use of laparoscopy to diagnose endometriosis (IETSI-2022 and SEF-2018) [23, 24], one guideline state a recommendation for either the use of diagnostic laparoscopy or medical empirical treatment (ESHRE-2022) [11], and one guideline state a recommendation in favor the use of laparoscopy to diagnose endometriosis (NICE-2017) [25]. The strength of recommendations was conditional in three guidelines (IETSI-2022, SEF-2018, and NICE-2017) [23–25] and not reported in one guideline (ESHRE-2022) [11]. Finally, the certainty of the evidence for the recommendations was not reported in two guidelines, state as very low in one guideline (IETSI-2022) [23], and state as high in one guideline (SEF-2018) [24]. **Table 3** details characteristics and methodological steps for each guideline.

## Generation of the clinical question generation and determination of the importance of outcomes

Two guides did not fully report information for all components of the PICO framework (SEF-2018 and NICE-2017) [24, 25]. The population of interest were reproductive-age women with suspected endometriosis for all guidelines. However, only one guideline specified details regarding the use of previous diagnostic tests and the type of endometriosis (IETSI-2022) [23]. Diagnostic laparoscopy was the intervention of interest in all guidelines but only one guideline specified the role of the test (IETSI-2022) [23]. Empirical medical treatment was the comparator in two guidelines (IETSI-2022 and ESHRE-2022) [11, 23]. Clinical benefits and complications or consequences of correct/incorrect classification were outcomes of interest in two guidelines (IETSI-2022 and ESHRE-2022) [11, 23], diagnostic test accuracy were outcomes of interest in two guidelines (IETSI-2022 and NICE-2017) [23, 25], and one guideline did not specify prioritized outcomes.

Regarding importance of outcomes, two guidelines did not report the importance of outcomes. The clinical patient important outcomes were at least important in one guideline (IETSI-2022) [23] and diagnostic test accuracy outcomes were critical in the remaining one (NICE-2017) [25].

## Systematic review of the evidence and certainty of the evidence assessment

All guidelines searched at least two databases and the date of search varied between 2016 to 2022. Three guidelines applied language restrictions, while this information was not reported by one guideline (SEF-2018) [24]. All CPG searched systematic reviews; three guidelines proposed searching for RCT; and one guideline prioritized searching for diagnostic accuracy observational studies (NICE-2017) [25]. All guidelines planned to use an instrument for risk of bias or study quality assessment, but only two guidelines specified QUADAS-2 (IETSI-2022

**Table 1. Characteristics of included clinical practice guidelines that assess the use of diagnostic laparoscopy for endometriosis.**

| Organization and year | Country | Perspective | Scope | Method to assess the certainty of the evidence | Classification and definition of the certainty of the evidence | System to classify the strength of recommendations | Classification and definition of the strength | Funding |
|---|---|---|---|---|---|---|---|---|
| Instituto de Evaluación de Tecnologías en Salud e Investigación, 2022 (IETSI-2022) | Peru | Population perspective | Diagnosis and treatment | GRADE | Certainty for each outcome:<br>• High: We are very confident that the true effect lies close to that of the estimate of the effect.<br>• Moderate: We are moderately confident in the effect estimate. The true effect is likely to be close to the estimate of the effect, but there is a possibility that it is substantially different.<br>• Low: Our confidence in the effect estimate is limited. The true effect may be substantially different from the estimate of the effect.<br>• Very low: We have very little confidence in the effect estimate. The true effect is likely to be substantially different from the estimate of effect.<br>Certainty for the recommendation:<br>• High: Our confidence in the evidence used to make the decision is high.<br>• Moderate: Our confidence in the evidence used to make the decision is moderate.<br>• Low: Our confidence in the evidence used to make the decision is low.<br>• Very low: Our confidence in the evidence used to make the decision is very low | GRADE | Conditional (the term 'We suggest' was used to phrase): This recommendation will be followed in most cases, although it might be appropriate not to apply it in some instances if this is justified.<br>Strong (the term 'We recommend' was used to phrase): This recommendation should be followed in all cases, except for specific and well-justified exceptions. | Institute of Health Technologies Assessment and Research of the Peruvian Social Security |
| European Society of Human Reproduction and Embryology, 2022 (ESHRE-2022) | European collaboration | Not reported | Primary prevention, diagnosis, and treatment | GRADE | High: We are very confident that the true effect lies close to that of the estimate of the effect.<br>Moderate: We are moderately confident in the effect estimate. The true effect is likely to be close to the estimate of the effect, but there is a possibility that it is substantially different.<br>Low: Our confidence in the effect estimate is limited. The true effect may be substantially different from the estimate of the effect.<br>Very low: We have very little confidence in the effect estimate. The true effect is likely to be substantially different from the estimate of effect. | GRADE | Weak* (the terms "It is conditionally recommended", "It is suggested", "Clinicians might", "Clinicians could consider", "Clinicians may/might consider" were used to phrase)<br>• Patients: Most people in your situation would want the recommended course of action but many would not<br>• Clinicians: Different choices will be appropriate for different patients and clinicians must make greater effort in helping patients to arrive at a decision consistence with their values and preference (shared decision-making)<br>• Policy: Policy making will require substantial debates and involvement of many stakeholders<br>Strong (the terms "Clinicians should", "It is recommended", "It is indicated", "Do" were used to phrase)<br>• Patients: Most people in your situation would want the recommended course of action and only a small proportion would not<br>• Clinicians: Most patients should receive the recommended course of action<br>• Policy: The recommendation can be adopted as a policy in most situation. | Not reported |

(*Continued*)

**Table 1.** (Continued)

| Organization and year | Country | Perspective | Scope | Method to assess the certainty of the evidence | Classification and definition of the certainty of the evidence | System to classify the strength of recommendations | Classification and definition of the strength | Funding |
|---|---|---|---|---|---|---|---|---|
| Sociedad Española de Fertilidad, 2018 (SEF-2018) | Spain | Not reported | Diagnosis and treatment | GRADE | High: High confidence that the effect estimator available in the scientific literature is very close to the true effect. Moderate: The effect estimator is likely to be close to the true effect, although there may be substantial differences. Low: The effect estimator may be substantially different from the true effect. Very low: It is very likely that the effect estimator is substantially different from the true effect. | GRADE | Weak* (the term 'We suggest' was used to phrase) • Patients: Many patients would want to follow the recommendation, but many would not. • Clinicians: Recognizing that different options will be appropriate for each patient should help them decide consistent with their values and preferences • Policy: Policy formulation will require considerable debate and involvement of diverse stakeholders Strong (the term 'We recommend' was used to phrase) • Patients: Most patients would agree with this. • Clinicians: Most patients should receive the procedure. adherence to it could be considered a quality criterion. • Policy: The recommendation can be adopted as a health policy measure in most recommendations. | Not reported. Acknowledges Theramex Laboratories for their support in logistical and methodological needs. |
| National Institute for Health and Care Excellence, 2017 (NICE-2017) | UK | | Diagnosis and treatment | GRADE | High: Further research is very unlikely to change our confidence in the estimate of effect. Moderate: Further research is likely to have an important impact on our confidence in the estimate of effect and may change the estimate Low: Further research is very likely to have an important impact on our confidence in the estimate of effect and is likely to change the estimate Very low: Any estimate of effect is very uncertain. | NICE | Must offer/refer/advice: Use the intervention. Consider: May use, depending on circumstances. So not offer/ do not refer/ do not advise; must not: Do not use the intervention. | National Institute for Health and Care Excellence |

* The GRADE methodology used the term "weak" previously, but currently, the term "conditional" is used.

**Abbreviations: UK:** United Kingdom; **GRADE:** Grading of Recommendations Assessment, Development and Evaluation; **NICE:** National Institute for Health and Care Excellence.

and NICE-2017) [23, 25]. Regarding the certainty of evidence assessment, all guidelines mentioned using the GRADE methodology.

## Preparation and report summaries of evidence tables with principal results

Only one guideline used SoF tables to report summaries of evidence and report the pre-test probability for the target condition (IETSI-2022) [23]. In general, guidelines mentioned a lack of studies on the clinical impact of using diagnostic laparoscopy. Sensitivity and specificity values were reported by three guidelines and showed slight variations (IETSI-2022, SEF-2018, and NICE-2017) [23–25] ranging from 0.94 to 0.98 for sensitivity and 0.79 for specificity. However, only one guideline explicitly addressed the clinical consequences of correct/incorrect classification with diagnostic laparoscopy and express it in number of cases per 100 individuals (IETSI-2022) [23]. In addition, two guidelines report the incidence of complications of the procedure, ranging from 2.3% to 6.8% (IETSI-2022 and ESHRE-2022) [11, 23].

Table 2. Quality appraisal of clinical practice guidelines that assess the use of diagnostic laparoscopy for endometriosis using the AGREE-II instrument.

| Domain | IETSI-2022 | ESHRE-2022 | SEF-2018 | NICE-2017 |
|---|---|---|---|---|
| Domain 1—Scope and Purpose | 88.9% | 77.8% | 68.5% | 90.9% |
| Domain 2—Stakeholder Involvement | 74.1% | 79.6% | 68.5% | 74.1% |
| Domain 3—Rigor of Development | 91.0% | 74.3% | 66.7% | 86.1% |
| Domain 4—Clarity of Presentation | 87.0% | 77.8% | 83.3% | 90.7% |
| Domain 5—Applicability | 50.0% | 38.9% | 36.1% | 86.1% |
| Domain 6—Editorial Independence | 83.3% | 77.8% | 27.8% | 83.3% |

**Abbreviations: AGREE-II:** Appraisal of guidelines research and evaluation—II; **IETSI-2022:** "Guía de práctica clínica para el diagnóstico y tratamiento de la endometriosis sintomática en mujeres en edad reproductiva" developed by IETSI in 2022; **ESHRE-2022:** "Endometriosis" developed by European Society of Human Reproduction and Embryology in 2022; **SEF-2018:** "Manejo de la paciente con endometriosis en la edad fértil" developed by Sociedad Española de Fertilidad in 2018; and **NICE-2017:** "Endometriosis: diagnosis and management" developed by NICE in 2017.

**Note:** With AGREE-II, global score for each domain ranging from 0 to 100%, where higher score indicates better quality. We considered that a GPC had an adequate quality when all six domains had a score ≥ 60%, and an adequate methodological quality when the third domain (rigor of development) had a score ≥ 60%.

The certainty of the evidence was appropriately expressed using the GRADE terminology in only one guideline and stated as low to very low certainty of the evidence (IETSI-2022) [23]. The remaining guidelines highlighted study limitations based on the risk of bias, using terms like "methodological limitations" or "high risk of bias", or based on the indirectness of the evidence, but did not detail the assessment of the remaining criteria proposed by GRADE.

## Discussion regarding evidence to decision criteria and reporting using a EtD table

Only one guideline used the EtD table to report the discussion regarding evidence to decision criteria and explicitly report the judgement for each criterion as proposed by GRADE (IETSI-2022) [23].

Regarding benefits, two guidelines, IETSI-2022 and ESHRE-2022, focused on potential clinical effects [11,23], one on laparoscopy's diagnostic accuracy (NICE-2017) [25], and the last, SEF-2018 [24], did not discuss benefits. IETSI-2022 considered benefits trivial, emphasizing empirical therapy's comparable advantages for high pre-test probability cases. ESHRE-2022 highlighted a lack of evidence for laparoscopy's superiority over empirical treatment.

Regarding harms, two guidelines discussed the consequences of incorrect classification or potential complications of laparoscopy concerning harms (IETSI-2022 and ESHRE-2022) [11,23], while others did not address this criterion. IETSI-2022 found harms to be trivial due to infrequent and not severe consequences, and ESHRE-2022 [11] highlighted the need to consider potential risks without explicitly judging this criterion.

Regarding the overall certainty of the evidence and the balance between benefits and harms, only one guideline reported the judgment for this criterion using the terminology proposed by GRADE (IETSI-2022) [23]. For IETSI-2022 [23] guideline, the overall certainty of the evidence was very low, and the balance possibly does not favor either the use of diagnostic laparoscopy or the empirical medical treatment in based of previous judgements for benefits, harms, and the certainty of the evidence. Similarly, ESHRE-2022 [11] guideline mention that there is no evidence of superiority of either approach and psychological benefits of a confirmed diagnosis should be weight against the value and risk of diagnostic laparoscopy. In contrast, SEF-2018 [24] guideline mention that the evidence on diagnostic accuracy of laparoscopy for endometriosis had limitations and there are empirically effective and safe treatments for symptom management. Finally, NICE-2017 [25] did not report discussions for this criterion.

**Table 3. Characteristics of recommendations and methodological steps to assess diagnostic test questions (use of laparoscopy as a diagnostic test for endometriosis) in included clinical practice guidelines.**

| Methodological steps to reach recommendations | | Clinical practice guidelines included | | | |
|---|---|---|---|---|---|
| | | **IETSI-2022** | **ESHRE-2022** | **SEF-2018** | **NICE-2017** |
| Generate the clinical question and decide on the importance of outcomes | Clinical question | In women with negative results on non-invasive imaging tests but with a high clinical suspicion of endometriosis, should laparoscopy be performed only for the disease diagnosis? | Does diagnostic laparoscopy compared to empirical medical treatment result in better symptom management in women suspected of endometriosis? | Should a laparoscopy be performed for diagnostic confirmation of possible endometriosis? | What is the accuracy of surgery with or without histological confirmation in diagnosing endometriosis? |
| | PICO framework | P: Women with negative results in non-invasive imaging tests with a high clinical suspicion of superficial or deep endometriosis and who do not have an immediate desire for fertility. I: Diagnostic laparoscopy. C: No diagnostic laparoscopy. Offer empirical medical treatment. O: Pain, fertility, dysmenorrhea, dyspareunia, complications (surgical wound infection, vascular, visceral, or solid organ injury), and diagnostic accuracy outcomes (sensitivity, specificity to detect endometriosis). | P: Women suspected of endometriosis I: Diagnostic laparoscopy. C: Empirical medical treatment. O: symptoms (pain) and complications. | P: Women on reproductive age with suspected endometriosis. I: Diagnostic laparoscopy C: Not reported. O: Not reported. | P: Symptomatic and asymptomatic women with suspected endometriosis. I: Diagnostic laparoscopy C: Not reported. O: Diagnostic accuracy outcomes (sensitivity, specificity, area under the curve) |
| | Role of the diagnostic test | Add-on | Not reported | Not reported | Not reported |
| | Assess patient important outcomes or consequences of correct/incorrect classification | Yes | Yes | No | No |
| | Importance of outcomes | Clinical outcomes were at least important, and diagnostic accuracy outcomes as surrogates. | Not reported. | Not reported | Diagnostic accuracy outcomes were critical |
| Perform the systematic review of the evidence and determine the certainty of the evidence | Databases and date of search | MEDLINE/PubMed and Cochrane library in February 2022. | MEDLINE/PubMed and Cochrane library in December 2020. | MEDLINE/PubMed, Cochrane library, EMBASE, Web of Science, and ClinicalTrials.gov in March 2018. | MEDLINE/PubMed, Cochrane library, CDSR, DARE, HTA, and EMBASE in December 2016. |
| | Restrictions | Only studies in English or Spanish language. | Only studies in English language. | Not reported. | Only studies in English language. |
| | Type of study to be search | SR of RCT. In absence, SR of diagnostic test observational studies. | SR. In absence, RCT, cohort studies and case reports according to the hierarchy of the levels of evidence. | SR. In absence, RCT, cohort studies and case reports according to the hierarchy of the levels of evidence. | SR, cross sectional studies, and cohort studies. |
| | Tool to assess risk of bias or quality of studies | AMSTAR-II, RoB 1.0, and QUADAS-2 for SR, RCT, and diagnostic test studies, respectively. | AMSTAR-II and ESHRE checklists for primary studies that assess selection, performance, attrition, detection, and other source of bias. | Not reported for SR. RoB 1.0 for RCT and other criteria proposed by Cochrane to assess bias in non-randomized studies. | AMSTAR-II and QUADAS-2 for SR and diagnostic test studies, respectively. |
| | Method or criteria to determine certainty of evidence | GRADE | GRADE | GRADE | GRADE |
| Prepare and report summaries of evidence tables | Type of evidence table | SoF table | Mention by the CPG as "not applicable" | Not reported | Mention by the CPG as "not applicable" |
| | Pre-test probability | 70% | Not reported | Not reported | Not reported |
| | Principal results | No SR of RCT were found. No data for pain, fertility, dysmenorrhea, or dyspareunia was found. Sensitivity (IC 95%): 0.98 (0.95 to 0.99). Diagnostic laparoscopy may classify 67 and 1 patients as TP and FN per 100, respectively (low certainty). Specificity (IC 95%): 0.79 (0.76 to 0.82). Diagnostic laparoscopy may classify 24 and 6 patients as TN and FP per 100, respectively (low certainty). The incidence of complications was 2.3% (very low certainty). | No studies comparing diagnostic laparoscopy and empirical medical treatment and their impact on symptoms was found. Overall incidence of complications was 6.8% in women of reproductive age with rectovaginal endometriosis in which laparoscopic excision was performed and 0.5% in patients in which diagnostic and operative laparoscopic surgeries for different type of pathologies was performed. | Sensitivity (IC 95%): 0.94 (0.80 to 0.98); and Specificity (IC 95%): 0.79 (0.67 to 0.87). However, studies that provide evidence for accuracy test outcomes had important methodological limitations and include mixed population. The histology did not confirm the diagnosis of endometriosis in 20% of cases with interobserver agreement and the studies have important limitations. Variations in sensitivity and specificity was reported based on the location and morphology of the lesions. | Not all diagnostic test studies included reported both sensitivity and specificity. In addition, risk of bias ranged from very high to moderate risk. Sensitivity (IC 95%) reported by two studies: 0.98 (0.95 to 0.99) and 0.97 (0.90 to 1). Specificity (IC 95%) reported by two studies: 0.79 (0.76 to 0.82) and 0.77 (0.72 to 0.82). |

(*Continued*)

**Table 3.** (Continued)

| Methodological steps to reach recommendations | | Clinical practice guidelines included | | | |
|---|---|---|---|---|---|
| | | IETSI-2022 | ESHRE-2022 | SEF-2018 | NICE-2017 |
| Discuss evidence to decision criteria | Use a EtD table | Yes | Yes (modified) | No | No |
| | Benefits | Trivial. Avoiding unnecessary medical treatment as consequence of correct classification may not yield substantial benefits in women with high pre-test probability of endometriosis, as adverse treatment events are infrequent, and if they occur, they are manageable. | Judgement not reported. No direct evidence was found to determine if diagnostic laparoscopy and further endometriosis treatment is better compared to empirical medical treatment for suspected endometriosis. | Judgement not reported. | Judgement not reported. Diagnosis of endometriosis is made based on visualization during laparoscopy. In addition, biopsies and histology may be important to diagnose other conditions and/or malignancies. Also, Committee considered that diagnosis would be dependent on the individual histopathologist. |
| | Harms | Trivial. The consequences of not providing early medical treatment due to misclassification and the complications of laparoscopy could be infrequent and potentially manageable. | Judgement not reported. No direct evidence was found. Consider possible risks of surgery. | Judgement not reported. | Judgement not reported. |
| | Certainty of the evidence (overall) | Very low | Judgement not reported. | Judgement not reported. | Judgement not reported. The risk of bias was very high to moderate. |
| | Values and preferences | No data for all critical outcomes. No evidence was found for all patient important outcomes so that they can be aware of potential clinical benefits and assess them. | Judgement not reported. Proofs of lesions could potentially be important for woman who have been suffering from the symptoms. | Judgement not reported. | Judgement not reported. Sensitivity and specificity were prioritized as critical outcomes and considered a proxy for patient level outcomes. |
| | Balance between benefits and harms | Possibly does not favor either the intervention or the comparison in base of trivial benefits and harms with very low certainty of evidence. | Judgement not reported. There is no evidence of superiority of either approach for any outcome. The psychosocial benefit of a confirmed diagnosis for an individual patient should be weight against the value and risk of laparoscopic surgery. | Judgement not reported. Lack of high-quality studies prevents issuing recommendations on who might benefit from laparoscopy for the diagnosis of endometriosis. The evidence on diagnostic accuracy has limitations, and diagnostic accuracy may vary depending on the lesion's location and morphology. Lesions may not always be detected, and their presence does not necessarily indicate the cause of symptoms such as pain or infertility. In contrast, there are empirically effective and safe treatments for symptom management. | Judgement not reported. |
| | Resource use or cost-effectiveness | Moderate costs for Peruvian social security ($13,362 per 100 diagnostic laparoscopies). | Judgement not reported. Mention by the CPG as expensive. | Judgement not reported. | Judgement not reported. A de novo economic model was conducted due lack of publish evidence. The economic model assume that laparoscopy is the "gold standard". However, this assumption is varied in sensitivity analysis. In based on the assumption described, laparoscopy is a highly effective form of diagnostic and the Committee believed most of the time a diagnostic laparoscopy is conducted it would have some clinical benefit. Nevertheless, when taken as one of many possible diagnosis/treatment strategies, surgical diagnosis is not preferred to empirical diagnosis and cheap treatment. |
| | Equity | Probably reduced. Probably not all health centers have the necessary human and logistical resources to provide diagnostic laparoscopies promptly. In consequence, the initiation of treatment may be delayed. | Judgement not reported. | Judgement not reported. | Judgement not reported. |
| | Acceptability | Varies. | Judgement not reported. Mention by the CPG as the laparoscopic surgery is widespread use. However, invasive and associated with morbidity and mortality. In contrast, empirical treatment without confirmed diagnosis is generally applied. | Judgement not reported. | Judgement not reported. |
| | Feasibility | Probably yes. Social security provides surgical rooms, laparoscopes, and human resources in specialized care centers. However, in some centers, laparoscopes are scarce, and it is necessary to assess whether the healthcare personnel are trained to perform the procedure. | Judgement not reported. | Judgement not reported. | Judgement not reported. |

*(Continued)*

**Table 3.** (Continued)

| Methodological steps to reach recommendations | | Clinical practice guidelines included | | | |
|---|---|---|---|---|---|
| | | **IETSI-2022** | **ESHRE-2022** | **SEF-2018** | **NICE-2017** |
| Formulate recommendations | Recommendation (as state in the CPG) | In women with negative results in non-invasive imaging tests but with persistent high clinical suspicion of endometriosis and no immediate fertility desire, we suggest not performing laparoscopy solely for diagnostic purposes of the disease | Both diagnostic laparoscopy and imaging combined with empirical treatment (oral contraceptive pill or progestogens) can be considered in women suspected of endometriosis. There is no evidence of superiority of either approach and pros and cons should be discussed with the patient. | Laparoscopy should not be performed for the diagnostic confirmation of possible endometriosis. Beyond the diagnostic validity of the test, this recommendation is based on the low probability that the result would alter the choice of empirical treatment that would be recommended in the absence of the test | Consider laparoscopy to diagnose endometriosis in women with suspected endometriosis, even if the ultrasound was normal |
| | Direction and strength | Against the diagnostic test. Conditional. | Recommendation for either the diagnostic test or the comparison. Strength not reported. | Against the diagnostic test. Conditional. | In favor of the diagnostic test. Conditional. |
| | Certainty of evidence for the recommendation | Very low. | Not reported. | High. * | Not reported. |

*The certainty of evidence for the recommendation was reported in this table as mentioned in the clinical practice guideline (⊕⊕⊕⊕ = high certainty).

**Abbreviations: IETSI-2022:** "Guía de práctica clínica para el diagnóstico y tratamiento de la endometriosis sintomática en mujeres en edad reproductiva" developed by IETSI in 2022; **ESHRE-2022:** "Endometriosis" developed by European Society of Human Reproduction and Embryology in 2022; **SEF-2018:** "Manejo de la paciente con endometriosis en la edad fértil" developed by Sociedad Española de Fertilidad in 2018; and **NICE-2017:** "Endometriosis: diagnosis and management" developed by NICE in 2017; **CPG:** Clinical practice guideline; **SR:** Systematic review; **RCT:** Randomized clinical trial, **AMSTAR-II:** A MeaSurement Tool to Assess systematic Reviews—II; **RoB:** Risk of bias tool; **QUADAS-2:** A Revised tool for the Quality Assessment of Diagnostic Accuracy Studies; **Sof:** Summary of findings; **EtD:** Evidence to decision; **GRADE:** Grading of Recommendations Assessment, Development and Evaluation.

Regarding resource use or cost-effectiveness, only one guideline reports the judgement for this criterion using the terminology proposed by GRADE (IETSI-2022) [23]. The IETSI-2022 [23] guideline, assess the costs associated with the use of diagnostic laparoscopy and concluded moderate costs. The ESHRE-2022 [11] guidelines classified the diagnostic laparoscopy as "expensive" and NICE-2017 [25] guidelines conducted an economic model, considering the limitations of assuming that diagnostic laparoscopy is the "gold standard", and concluded that is a cost-effective form of diagnosis. Nevertheless, mention that when taken as one of many possible diagnosis/treatment strategies, surgical diagnosis is not preferred to empirical diagnosis and not expensive treatment. In addition, SEF-2018 [24] guideline did not report discussion for this criterion.

Regarding the remaining criteria, patient values and preferences were discussed by three guidelines (IETSI-2022, ESHRE-2022, and NICE-2017) [11,23,25], the impact on health equity was discussed by one guideline (probably reduced) (IETSI-2022) [23], the acceptability of laparoscopy was discussed by two guidelines (varies) (IETSI-2022 and ESHRE-2022) [23,25], and feasibility was discussed by one guideline (likely feasible in specialized centers but may not be available in other regions in the short term) (IETSI-2022) [23].

## Discussion

In our analysis, despite a similar body of evidence on laparoscopy's diagnostic accuracy, guidelines exhibited divergent recommendations. Variability in prioritizing outcomes, especially clinical benefits or consequences on patient-important outcomes, may contribute to these differences. While there's a proposed need for Clinical Practice Guidelines (CPGs) to evaluate the clinical impact of a test, our review indicates that this approach is not commonly practiced. A study analyzing 15 guidelines found that out of 15 recommendations on diagnostic test use, only 3 considered the effectiveness of treatments following the tests, and none linked the diagnostic test results with the decision-making process for subsequent treatment administration [9].

Guideline recommendations vary due to inconsistent consideration of criteria in transitioning from evidence to decision and variability in how guidelines address and weigh these criteria. The balance between benefits and harms, along with evidence certainty, is crucial in shaping the initial recommendation direction and is often considered in guidelines [26]. In our results, only one guideline comprehensively assessed the potential benefits by evaluating the absolute effects of accurate laparoscopy classification in women with a clearly defined high pre-test probability. This guideline also effectively communicated the certainty of the evidence for these outcomes, an important aspect considering that the magnitude of benefits can varied according to the initial probability of the condition [15]. Furthermore, only two guidelines addressed potential harms. This lack of emphasis on the burden of diagnostic tests and the methodological shortcomings in gathering such information have been previously noted. A study highlighted that only 3 of 15 guidelines reviewed considered the harm burden related to diagnostic tests, and this consideration was not supported by systematic searches for evidence [9].

The balance of effects and certainty of evidence results must be weighed against other EtD framework criteria for the final recommendation. Considerations on resource use or cost-effectiveness, crucial in decision-making, vary based on contextual factors [27]. In this regard, our analysis found that guidelines assigned different importance to resource use in issuing their recommendations. The variability in the importance given by the guidelines to the discussion of this criterion could be explained by the perspective of the guidelines. For instance, for guidelines with a population perspective, resource use must justify the magnitude of the benefits found to issue a favorable recommendation; otherwise, it is more likely to issue a recommendation against it, as reported by the IETSI-2022 guideline. The limited information on the perspective of the included guidelines does not allow for a more detailed understanding of the importance assigned to each criterion. This lack of information aligns with findings from a study in which only 24% of guidelines from various health areas reported the perspective to be used [26].

All guidelines used GRADE to assess evidence certainty and issue recommendations. However, only one complied with reporting and explaining certainty in detail, using GRADE's recommended phrasing and terminology. Utilizing SoF tables aids quick and improved interpretation for decision-making [28]. Furthermore, the EtD table is one of the most frequently used decision frameworks [26], and its use allows guiding decision-making through multiple criteria in a systematic, explicit, and transparent way to issue recommendations, stating their direction and strength [29]. In the evaluation of EtD criteria within clinical guidelines, equity, acceptability, and feasibility were the least addressed, aligning with a study that found these criteria discussed in only 16%, 28%, and 34% of 68 guidelines, respectively [26].

Despite adequate overall methodological quality, not all guidelines consistently applied GRADE methodology. The most GRADE-adherent guideline scored highest in the third domain. This aligns with a study on 15 Australian guidelines, indicating inconsistent GRADE application (73.3%). However, those closely following GRADE achieved higher methodological rigor scores. The authors noted that 40% of the guidelines failed to report sufficient information on the importance of outcomes, 13% did not use GRADE's evidence certainty categories, 47% did not present results using evidence summary tables, and 80% did not present the EtD framework. Challenges in applying GRADE to assess clinical impact of diagnostic tests include skepticism on recommendation strength, a preconceived confidence in benefits despite low evidence certainty, and limitations in guideline development groups' understanding of the methodology [30]. However, published case analyses on diagnostic test questions can guide these groups [31]. Additionally, the GRADE series offers several articles clarifying the steps to make recommendations on diagnostic tests and there exists publications about

how to use the EtD frameworks [32]. More articles are being published to operationalize the evaluation of criteria for making a recommendation, which improve the interrater agreement for making a for/against recommendation [33]. The analysis table we present in this article is also available for use as a tool to verify the report and analyze the consistency of the recommendation on the use of a diagnostic test. Finally, educational programs are available on the GRADE Working Group website (https://www.gradeworkinggroup.org/) to enhance the training of guideline development groups in using these methods and improving their understanding.

Despite the above, although laparoscopy reports good diagnostic and therapeutic utility in the clinical recommendations, some endometriotic lesions may be missed because some implants may be very small or hidden to white light laparoscopy, so near infrared radiation (NIR) intraoperative imaging technology has shown to be a useful tool in these situations [34]. In addition to this, it has been reported that a miRNA pattern could be a promising and cost-effective tool for the diagnosis of endometriotic implants in patients with or without symptoms, mainly by liquid biopsy [35]. Therefore, these considerations should be taken into account in the development of future clinical recommendations for patients with endometriosis.

## Strengths and limitations

This study analyzed four CPGs based on methodology indicators like recommendation characteristics, diagnostic test question approach, PICO structure, evidence search, limitations assessment, evidence synthesis, and transition to recommendation. The AGREE instrument guided the appraisal. Limitations include the inability to access dynamic panel information, explaining decisions and transparency gaps due to document analysis.

## Conclusions

This scoping review highlights the variability in recommendations for diagnostic laparoscopy in endometriosis among different CPGs. Despite similar evidence on diagnostic accuracy, discrepancies stem from differing prioritizations of clinical impact assessment and methodological approaches. There's a need for standardized and transparent guideline development processes, especially addressing the clinical utility of diagnostic tests.

## Supporting information

**S1 Table. PRISMA extension for Scoping Reviews (PRISMA-ScR) checklist.**
(DOCX)

**S2 Table. Search strategy.**
(DOCX)

**S3 Table. Excluded studies.**
(DOCX)

## Author Contributions

**Conceptualization:** Frank Zela-Coila, Carlos Quispe-Vicuña, Janeth N. Nuñez-Lupaca, Milagros Aparicio-Curazi, Sergio Goicochea-Lugo.

**Data curation:** Frank Zela-Coila, Carlos Quispe-Vicuña, Janeth N. Nuñez-Lupaca, Milagros Aparicio-Curazi, Sergio Goicochea-Lugo.

**Formal analysis:** Frank Zela-Coila, Carlos Quispe-Vicuña, Janeth N. Nuñez-Lupaca, Milagros Aparicio-Curazi, Sergio Goicochea-Lugo.

**Investigation:** Frank Zela-Coila, Carlos Quispe-Vicuña, Janeth N. Nuñez-Lupaca, Milagros Aparicio-Curazi, Sergio Goicochea-Lugo.

**Methodology:** Frank Zela-Coila, Carlos Quispe-Vicuña, Janeth N. Nuñez-Lupaca, Milagros Aparicio-Curazi, Sergio Goicochea-Lugo.

**Project administration:** Frank Zela-Coila, Carlos Quispe-Vicuña, Janeth N. Nuñez-Lupaca, Milagros Aparicio-Curazi, Sergio Goicochea-Lugo.

**Resources:** Frank Zela-Coila, Carlos Quispe-Vicuña, Janeth N. Nuñez-Lupaca, Milagros Aparicio-Curazi, Sergio Goicochea-Lugo.

**Software:** Frank Zela-Coila, Carlos Quispe-Vicuña, Janeth N. Nuñez-Lupaca, Milagros Aparicio-Curazi, Sergio Goicochea-Lugo.

**Supervision:** Frank Zela-Coila, Carlos Quispe-Vicuña, Janeth N. Nuñez-Lupaca, Milagros Aparicio-Curazi, Sergio Goicochea-Lugo.

**Validation:** Frank Zela-Coila, Carlos Quispe-Vicuña, Janeth N. Nuñez-Lupaca, Milagros Aparicio-Curazi, Sergio Goicochea-Lugo.

**Writing – original draft:** Frank Zela-Coila, Carlos Quispe-Vicuña, Janeth N. Nuñez-Lupaca, Milagros Aparicio-Curazi, Sergio Goicochea-Lugo.

**Writing – review & editing:** Frank Zela-Coila, Carlos Quispe-Vicuña, Janeth N. Nuñez-Lupaca, Milagros Aparicio-Curazi, Sergio Goicochea-Lugo.

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
