## [Decision Letter · Decision Letter 0]

12 Jul 2024

PONE-D-24-16093Comparison of clinical practice guidelines methods to reach diagnostic test recommendations regarding diagnostic laparoscopy for endometriosis: a scoping review.PLOS ONE

Dear Dr. Goicochea-Lugo,

Thank you for submitting your manuscript to PLOS ONE. After careful consideration, we feel that it has merit but does not fully meet PLOS ONE’s publication criteria as it currently stands. Therefore, we invite you to submit a revised version of the manuscript that addresses the points raised during the review process.

We look forward to receiving your revised manuscript.

Kind regards,

Diego Raimondo

Academic Editor

PLOS ONE

Journal Requirements:

2. We note that this manuscript is a systematic review or meta-analysis; our author guidelines therefore require that you use PRISMA guidance to help improve reporting quality of this type of study. Please upload copies of the completed PRISMA checklist as Supporting Information with a file name “PRISMA checklist”.

Reviewers' comments:

Reviewer's Responses to Questions

**Comments to the Author**

1. Is the manuscript technically sound, and do the data support the conclusions?

Reviewer #1: Yes

Reviewer #2: Yes

2. Has the statistical analysis been performed appropriately and rigorously? 

Reviewer #1: N/A

Reviewer #2: I Don't Know

3. Have the authors made all data underlying the findings in their manuscript fully available?

Reviewer #1: No

Reviewer #2: Yes

4. Is the manuscript presented in an intelligible fashion and written in standard English?

Reviewer #1: Yes

Reviewer #2: Yes

5. Review Comments to the Author

Reviewer #1: I read with great interest the present Manuscript which falls within the aim of the Journal. In my honest opinion, the topic is interesting enough to attract the readers’ attention. Methodology is accurate and conclusions are supported by the data analysis. Nevertheless, authors should clarify some points and improve the discussion citing relevant and novel key articles about the topic. For all those reasons, I suggested performing the minor revisions.

- Line 53: edit, and write “to keep in mind”;

- Inclusion and exclusion criteria should be clarified in Methods. For example, Authors may specify whether they included abstract-only publications, Systematic Reviews and Meta-Analyses;

- In Discussion, Authors should parallel the use of Laparoscopy both in diagnosis and treatment with the benefits of sclerotherapy in preserving ovarian parenchyma. Please consider: “Ronsini C, Fumiento P, Iavarone I, Greco PF, Cobellis L, De Franciscis P. Liquid Biopsy in Endometriosis: A Systematic Review. Int J Mol Sci. 2023;24(7):6116. Published 2023 Mar 24. doi:10.3390/ijms24076116”;

- Line 136: edit, and write “was found”;

- Please provide Figure 1 in better quality. It appears blurry.

Reviewer #2: thank you for giving me the chance to review this interesting review highlighting the variability in recommendations for diagnostic laparoscopy in endometriosis among different CPGs.

Despite good overall merit, I have some minor comments:

- please correct some typos thorughout the text (example EMABSE)

- discussing on clinical benefit of diagnostic laparoscopy one may argue that some occult disease can be omitted at white light laparoscopy or deep lesion in the parametrial area (doi 10.1111/aogs.13866)

6. PLOS authors have the option to publish the peer review history of their article (what does this mean?). If published, this will include your full peer review and any attached files.

Reviewer #1: **Yes: **Carlo Ronsini

Reviewer #2: No

---

## [Author Response · Author response to Decision Letter 0]

9 Aug 2024

Response to Reviewer 1

1. Line 53: edit, and write “to keep in mind”

We appreciate your comment and accept the suggestion. In line 53, we accepted the change.

2. Inclusion and exclusion criteria should be clarified in Methods. For example, Authors may specify whether they included abstract-only publications, Systematic Reviews and Meta-AnalysesWe thank you for your comment. We made the change in the manuscript. We specify that we include only clinical practice guidelines and exclude systematic reviews, observational studies, clinical trials, conference proceedings, letters to the editor, and case reports.

3. - In Discussion, Authors should parallel the use of Laparoscopy both in diagnosis and treatment with the benefits of sclerotherapy in preserving ovarian parenchyma. Please consider: “Ronsini C, Fumiento P, Iavarone I, Greco PF, Cobellis L, De Franciscis P. Liquid Biopsy in Endometriosis: A Systematic Review. Int J Mol Sci. 2023;24(7):6116. Published 2023 Mar 24. doi:10.3390/ijms24076116” 

We appreciate your comment. We added the following text in discussion: “Despite the above, although laparoscopy reports good diagnostic and therapeutic utility in the clinical recommendations, some endometriotic lesions may be missed because some implants may be very small or hidden to white light laparoscopy, so near infrared radiation (NIR) intraoperative imaging technology has proven to be a useful tool in these situations [34]. In addition to this, it has been reported that a miRNA pattern could be a promising and cost-effective tool for the diagnosis of endometriotic implants in patients with or without symptoms, mainly by liquid biopsy [35]. Therefore, these considerations should be taken into account in the development of future clinical recommendations for patients with endometriosis”.

4. Line 136: edit, and write “was found” We appreciate your comment and accept the suggestion.

In line 136, we accepted the change.

5. Please provide Figure 1 in better quality. It appears blurry. 

Thank you for your comment. The quality of Figure 1 was improved

Response to Reviewer 2

1. Please correct some typos thorughout the text (example EMABSE) 

We appreciate your comment. Errata in the text were corrected

2. Discussing on clinical benefit of diagnostic laparoscopy one may argue that some occult disease can be omitted at white light laparoscopy or deep lesion in the parametrial area (doi 10.1111/aogs.13866) 

We appreciate your comment. We added the following text in discussion: “Despite the above, although laparoscopy and histologic examination remain the gold standard for the diagnosis of endometriosis, some endometriotic lesions may be missed because some implants may be very small or hidden to white light laparoscopy, so near infrared radiation (NIR) intraoperative imaging technology has proven to be a useful tool in these situations”.

---

## [Editor Report · Decision Letter 1]

23 Aug 2024

Comparison of clinical practice guidelines methods to reach diagnostic test recommendations regarding diagnostic laparoscopy for endometriosis: a scoping review.

PONE-D-24-16093R1

Dear Dr. Goicochea-Lugo,

We’re pleased to inform you that your manuscript has been judged scientifically suitable for publication and will be formally accepted for publication once it meets all outstanding technical requirements.

Kind regards,

Diego Raimondo

Academic Editor

PLOS ONE

---

## [Editor Report · Acceptance letter]

6 Sep 2024

PONE-D-24-16093R1 

PLOS ONE

Dear Dr. Goicochea-Lugo, 

I'm pleased to inform you that your manuscript has been deemed suitable for publication in PLOS ONE. Congratulations! Your manuscript is now being handed over to our production team.

Kind regards, 

on behalf of

Dr. Diego Raimondo 

Academic Editor

PLOS ONE